# Engineering Properties of New-Age (Nano) Modified Emulsion (NME) Stabilised Naturally Available Granular Road Pavement Materials Explained Using Basic Chemistry

Gerrit J. Jordaan [1,2,*] and Wynand J. vdM. Steyn [3]

1   Department of Civil Engineering, University of Pretoria, Pretoria 0002, South Africa
2   Jordaan Professional Services (Pty) Ltd., Pretoria 0062, South Africa
3   School of Engineering, Department of Civil Engineering, University of Pretoria, Pretoria 0002, South Africa; Wynand.steyn@up.ac.za
*   Correspondence: jordaangj@tshepega.co.za; Tel.: +27-(0)-82-416-4945

**Featured Application:** The engineering properties obtainable during the construction of road pavement layers using reactive stabilising agents are traditionally determined using a trail-and-error approach based on material indicator tests, some developed more than a century ago. These tests inhibit the limitations associated with an empirically derived test or method, often leading to costly failures. The introduction of proven and applicable nanotechnologies to treat or stabilise granular material for use in pavement layers could result in the rejection of these technologies if not based on the understanding of the basic chemistry of both the stabilising agent and the mineralogy of the granular materials. This article gives a basic explanation of elementary chemistry that will affect the physical engineering properties (stresses and strains) that can be expected using available, proven and cost-effective nanotechnologies to improve granular materials for use in a pavement structure.

**Abstract:** Nanoscale organofunctional silanes have been developed, tested and successfully applied to protect stone buildings in Europe against climatic effects since the 1860s. The same nanotechnologies can also be used in pavement engineering to create strong chemical bonds between a stabilising agent and granular material. The attachment of the organofunctional silane to a material also changes the surface of the material to become hydrophobic, thereby considerably reducing future chemical weathering. These properties allow naturally available materials to be used in any pavement layer at a low risk. In the built environment, scientists soon determined that the successful use of an organo-silane depends on the type and condition of the stone to be treated. The same principles apply to the implementation of applicable nanotechnologies in pavement engineering. Understanding the basic chemistry, determining the properties of the stabilising agent and the organofunctional modifying agent and the chemical interaction with the primary and secondary minerals of the material are essential for the successful application of these technologies in pavement engineering. This paper explains some basic chemistry, which fundamentally influences engineering outputs that can be achieved using New-age (Nano) Modified Emulsions (NME) stabilising agents with naturally available granular materials in all road pavement layers below the surfacing.

**Keywords:** nanotechnology in pavement engineering; influence of emulsifying agents (surfactants); chemistry of surfactants in emulsions; stabilisations of naturally available granular materials; mineralogy compatible nano-modified emulsions; organofunctional silanes; anionic emulsions; cationic emulsions; new-age (nano) modified emulsions (NME)

## 1. Introduction

With the new millennium, the world also entered the fourth Industrial Revolution (4RI) [1] that will "broaden and deepen the connections between the biological, physical

and digital worlds in unprecedented ways" [2]. This phase of development will have an impact on all spheres of life, affecting all industries, bringing together multiple technologies in creating a new modern environment. Relatively new industries such as the Information Technology (IT) industry will experience the 4RI as a natural extension of ongoing developments. However, many "traditional" industries will experience the 4RI as disruptive [3], requiring an extensive change in "set in stone practices," together with a considerable change in the mindset of practitioners accustomed to traditionally operating procedures. The pavement engineering fraternity, with specific reference to the classification and use of naturally available granular materials, may well fit into the latter category. An understanding of the basic scientific principles will greatly assist in the general acceptance and implementation of these technologies in traditional industries.

Road pavement granular materials are traditionally classified using empirically derived indicator tests (some developed more than a hundred years ago). In contrast, pavement engineering design has evolved over the last few decades from empirically derived methods to a more fundamental scientific approach with the development and introduction into practice of Mechanistic–Empirical (ME) design methods. These methods use fundamental failure theories based on the computer modulation of road pavement structures to analyse and predict expected pavement behaviour trends based on calculated basic physics, i.e., stresses and strains within the different pavement layers. Unfortunately, the testing and characterisation of granular materials for use in pavement structures have not shown the same advances.

The successful introduction and general acceptance of cost-effective proven and available nanotechnologies to improve granular materials for use in road pavement structures will not succeed using empirical tests and methods, relying on a trial-and-error approach. A scientific understanding of the mineralogy of the granular materials as well as the basic chemistry and chemical reaction caused by the use of proven/available technologies in a traditionally conservative industry is required in line with the expected developments during the 4RI—the use and combination of available information and technologies to understand the basic science behind technologies and predict, with improved confidence, results. Such an understanding obviously also applies to traditionally used reactive stabilising agents (e.g., cement, lime, modified bitumen emulsions, etc.), the general use of which often results in "unexplained" costly failures. These traditionally used stabilising agents are each not a single product but contain various combinations of different chemical components, which could react differently with different materials. This article is limited to the basic understanding of the chemical interaction of organofunctional nano-silane modified emulsions in combination with granular material for use in pavement layers below the surfacing.

Nanotechnology products (nano-silanes) have been used in the built environment to improve, strengthen and protect stone buildings in Europe for almost 200 years [4]. The same technology could find immediate application in the field of pavement engineering to improve the cost-effective use of granular materials in road pavement structures. Scientists responsible for the development of the first nano-silane products in the built environment in the early 1800s [4–6] soon established that the type of stone (primary minerals) and the condition of the stone (presence and quantity of secondary minerals) considerably affected results achievable through any specific nano-silane treatment. These lessons learned from the built environment must form the basis for the successful introduction of these available, proven nanotechnologies in pavement engineering.

It follows that engineers need to have at least a basic understanding of the chemistry involved in the use of these materials. The informed selection of applicable material compatible nanotechnologies that are stable for practical application under often challenging practical field conditions will prove to be the difference between success and failure. It is not the intention to make pavement engineers chemists, but only to provide them with the necessary tools (basic knowledge) to make informed decisions in practice. The key is

not to confuse with fundamental, detailed chemistry but to provide practical clarifications, explaining results and showing the consequences of insufficient specifications.

Knowledge about these basic chemical concepts controlling material behaviour in the field will assist greatly in the informed application of materials and the use of material-compatible technologies by engineers specialising in road pavement engineering. The successful introduction of new-age nanotechnologies is disruptive to traditional pavement engineering practices, enabling the use of granular materials generally perceived as unsuitable in quality. Hence, the understanding of the basic chemistry controlling nanotechnology characteristics in an emulsion, in combination with the mineralogy of naturally available materials, can facilitate the acceptance of the engineering benefits in road construction, leading to a substantial reduction in transportation infrastructure costs [7–9]. This article will also demonstrate the importance of more specific specifications with regard to the chemical properties of the emulsifying agent (surfactant) as a nano-particle used in the manufacturing of a specific stabilising agent (bitumen emulsion), which forms the foundation for the production of a material compatible New-age (Nano) Modified Emulsion (NME). The chemical characteristics of both nano-particles, i.e., the surfactant and the nano-silane modifier, are important factors in determining the engineering properties to be achieved when applied in practice for the treatment/stabilisation of available granular materials.

## 2. Background

The use of nano-scale products for the stabilisation of granular materials in the construction of the pavement layers is nothing new. Per definition, products such as cement, lime as well as bitumen emulsion all incorporate nano-scale particles and can be considered as nanotechnology products, i.e., containing particles of which at least one dimension is between 1 and 100 nm in size [10]. The use of nanotechnology as a science only became of interest after the development of equipment enabling scientists (including chemists, physicians and engineers) to see and manipulate nano-scale particles at a molecular level in the 1980s/90s [11,12]. This ability to manipulate nano-scale products has had an impact on all industries, including the built environment where silicon-based nanotechnology products are being used to improve building materials across basically all spheres of activity. However, nano-silane products have been developed, tested and used in the built environment to protect stone buildings in Europe since the 1860's [4]. The more than 150 years of "lessons learnt" from the built environment can assist pavement engineers to fast-track the implementation of these proven technologies to also protect and improve naturally available granular materials for use in the design and construction of roads. Experience in southern Africa [13] has shown that considerable savings are a reality through the implementation of nano-silane technologies used for the treatment and stabilisation of granular materials for all layers below the surfacing.

Materials used in road pavements are traditionally classified using empirically derived criteria based on material indicator tests dating back more than a century [14]. These material classification systems often classify naturally available granular materials in climatic regions of the world associated with a high potential for chemical decomposition (high temperatures in combination with seasonal rainfall [15]) as "non-standard," "marginal" or even "sub-standard" [16]). Available and applicable nanotechnologies that could enable the use of these materials at a low-risk in pavement structures could substantially reduce the unit costs of road infrastructure, specifically in these regions.

Bitumen emulsion technology dates back to the early 1900s [17] when a nano-scale particle was discovered that enables an oil substance (e.g., bitumen) to be mixed with an aqueous substance (i.e., a water-based substance). This nano-scale particle is commonly referred to as an emulsifying agent (known in chemistry as a surfactant and in engineering practice commonly referred to as a "soap"). Bitumen emulsion technology enables a relatively low bitumen content to be mixed with granular materials at ambient temperatures to construct road pavement layers. Emulsion technology incorporates several advantages, including the ability to accommodate considerably higher tensile strains in comparison to the

unstabilised granular material as well as cement-bound materials [14,18]. However, similarly to the manufacturing of asphalt, it is still a requirement to use aggregate/stone/gravel granular materials of a relatively high quality together with the bitumen emulsion for the construction of a pavement layer meeting the required engineering specifications [19].

Similar to the manufacturing of asphalt for surfacings, the unmodified bitumen forms no chemical bonds with the granular aggregate/stone materials in the mix. Strength is only achieved through covalent bonds (relatively weak) and mechanical forces created through granular interlock and absorption of the bitumen into a porous surface of the aggregate [15]. For this reason, some aggregates containing a high silicon content, which usually result in relatively "clean breaks" during crushing, are notoriously difficult to use for the manufacturing of asphalt or bitumen emulsion stabilised layers meeting the engineering specifications. In asphalt and bitumen emulsion mixes, the use of materials conforming to specific grading envelopes are of major importance to create a firm granular matrix, which results in high interlocking mechanical forces being formed.

The ability to create strong chemical bonds between the stabilising agent (e.g., bitumen or equivalent polymer) and the granular material to be stabilised (aggregate/stone/soil) can be achieved through the introduction and use of proven material compatible organofunctional nano-silanes. These nano-silane products attach to the granular materials, creating relatively strong ionic-chemical bonds. The organofunctional part of the nano-silane particle is hydrophilic, rendering the surface of each granular particle of the material to become hydrophobic during consolidation, preventing water access to primary and secondarily minerals comprising each of the granular particles within the mix. The high chemical bond strengths and the enacted hydrophobicity enable materials classified as "non-standard," "marginal" or "sub-standard" to be utilised successfully within any pavement layer below the surfacing, at low risk. The introduction of material-compatible organofunctional nano-silanes in the field of pavement engineering is a typical example of a disruptive technology [3], requiring traditional approaches to the use of materials in pavement engineering to become irrelevant. This combination of existing technologies in combination with an improved scientific understanding and knowledge forms the cornerstone of the 4RI in traditional industries, especially with regard to the cost-effective provision of macro infrastructure projects using "smart" materials in a cost-effective manner. The general acceptance of such disruptive technologies will require the necessary improved knowledge of the basic supportive science to become everyday practice. The key is not to overwhelm the practicing pavement engineer with complex fundamental chemistry but to simplify facts to be easily understandable in support of practical implementation.

Many products have been introduced throughout the last few decades claiming to be able to provide the ability to improve granular material characteristics to enable the use thereof in road pavement structures. These so-called "snake-oils" have generally failed to meet expectations. In the absence of a scientifically-based approach to granular material investigations and tests indicative of engineering principles (e.g., stresses, strains and durability), the same can happen with the introduction of applicable/proven nano-silane technologies for the improvement/stabilisation of granular materials in pavement engineering [20].

The work performed by scientists in the built environment dating back almost 200 years established the basic requirements for the successful application of any specific organofunctional silane technology to improve granular/stone material characteristics. These requirements are fundamentally based on the compatibility with the type of stone (primary minerals) and condition of the stone (presence and amount of secondary minerals in the granular/stone that developed as a result of weathering due to chemical decomposition) [21,22]. It follows that material classification should, at least, include the scientific testing of the primary and secondary minerals present in the available granular materials [20]. Knowledge about the mineralogy of the materials and the environmental conditions favouring weathering through chemical decomposition [22] and the formation of secondary minerals in granular materials will enable engineers to select with confidence

a material compatible technology to enhance/stabilise marginal granular materials for use in all of the pavement layers below the surfacing.

The same principles apply to the characteristics of the nano-particles (surfactant) used to produce the emulsion stabilising agent (e.g., bitumen emulsion) and the effect thereof on the engineering properties when used in combination with an organofunctional silane modifying agent. In effect, the introduction of a material compatible nano-silane creates a modified emulsifying agent combining two different nano-particles. Each of these nano-particles could have a considerable influence on the engineering properties when used in the production of a modified emulsion stabilising agent and applied to specific granular material as a stabilising agent.

### 3. Traditional Use of Nanotechnology Products in Road Pavement Engineering

The use of nano-scale material in the road industry (over and above the use of lime and cement) as stabilisation agents dates back more than a century, with the development of bitumen emulsions in the early 1900s [8]. As per definition [17,23–25], bitumen emulsion consists of bitumen, water and an emulsifying agent. The emulsifying agent is, in fact, a nano-scale particle commonly referred to as a "Janus" particle [26] (from the Greek mythology meaning "two-faced") due to the dual nature of the emulsifying agent nano-particle, attracting oil on the one side and water on the other side.

It is not the intention to discuss in detail the technology involved in the production of bitumen emulsions [17,23,24,26,27]. It is generally known that numerous production factors, bitumen rheology, etc., could influence the characteristics of bitumen emulsion, which are discussed in detail in numerous publications. The objective of this paper is to concentrate on the effects of the chemistry involved in the additives and modifications used in the manufacturing of bitumen emulsion as a stabilising agent. Understanding the role of the emulsifying agent nano-particle is comparable to that of organofunctional nano-silane products used in the built environment. Hence, the discussion in this paper is limited to the role of the emulsifying agent that enables water to be mixed with oil (organic substance, in this case, bitumen) substances, which, under normal conditions, do not mix. Not only does the emulsifying agent enable the oil (e.g., bitumen) to be mixed with water, but crucially, when modified with the addition of a material compatible nano-silane, the characteristics could also dramatically influence the engineering properties achievable when the nano-modified emulsion is used with a specific granular material. It follows that a basic understanding of the role of the nano-emulsifying agent also needs to be addressed.

The mixing of oil (e.g., bitumen) and water is achieved through the addition of a chemical nano-particle (the emulsifying agent, soap or surfactant) at high shear (e.g., high mix revolutions), which forces the oil and water together with the emulsifying agent through small openings or between plates, which enables the bitumen (oil) particles to be separated and mixed and attached to the emulsifying agent already mixed and attached to the water molecules. The emulsifying agent typically has a hydrophilic (water-loving) head and a lipophilic (oil-loving) (hydrophobic) tail consisting of between 12 and 18 (or even more) carbon atoms [17]. The chemical composition of typical emulsifying agents is shown in Figure 1 (anionic) and Figure 2 (cationic). The hydrocarbon tail of the emulsifying agents is often replaced in chemical formulas by the letter "R".

The higher the number of carbon atoms in the hydrocarbon tail, the more firmly the emulsifying agent will attach to the organic stabilising agent (e.g., bitumen molecule). The hydrocarbon tail embeds itself into the bitumen molecule. In comparison, if the earth resembles the size of a bitumen particle, the hydrocarbon tail of a good emulsifying agent (high number of carbon atoms) will typically penetrate the crust of the earth to a depth of approximately 8 km (equivalent to approximately 5 nm in a bitumen molecule) and covers an area of approximately 10 km$^2$ [25].



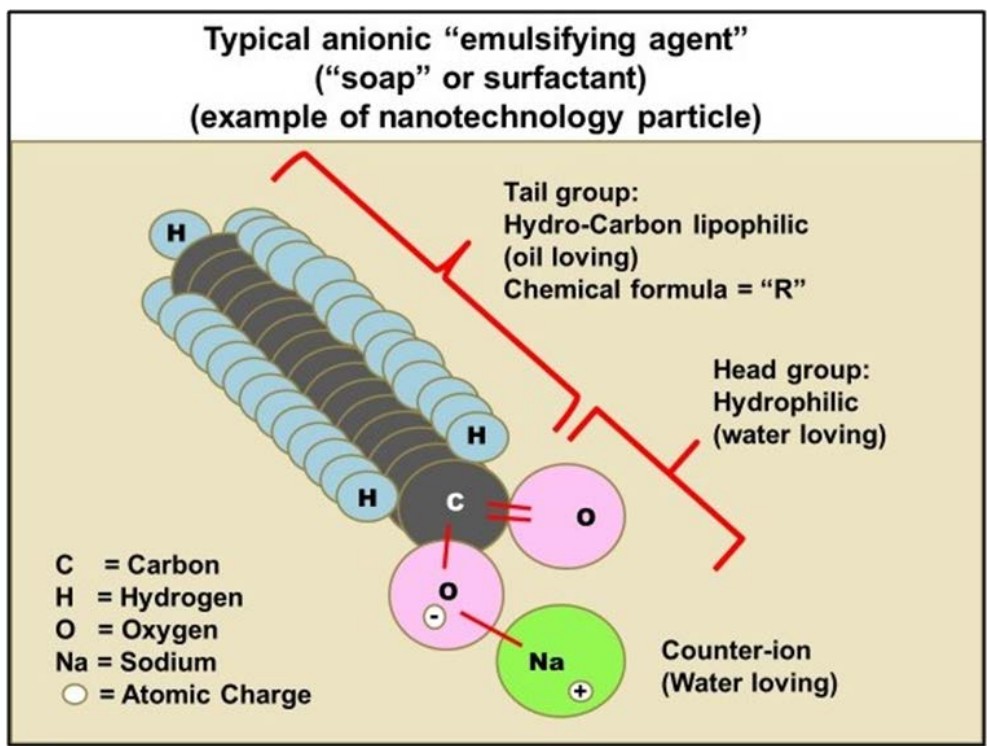

**Figure 1.** Typical composition of an anionic emulsifying agent.

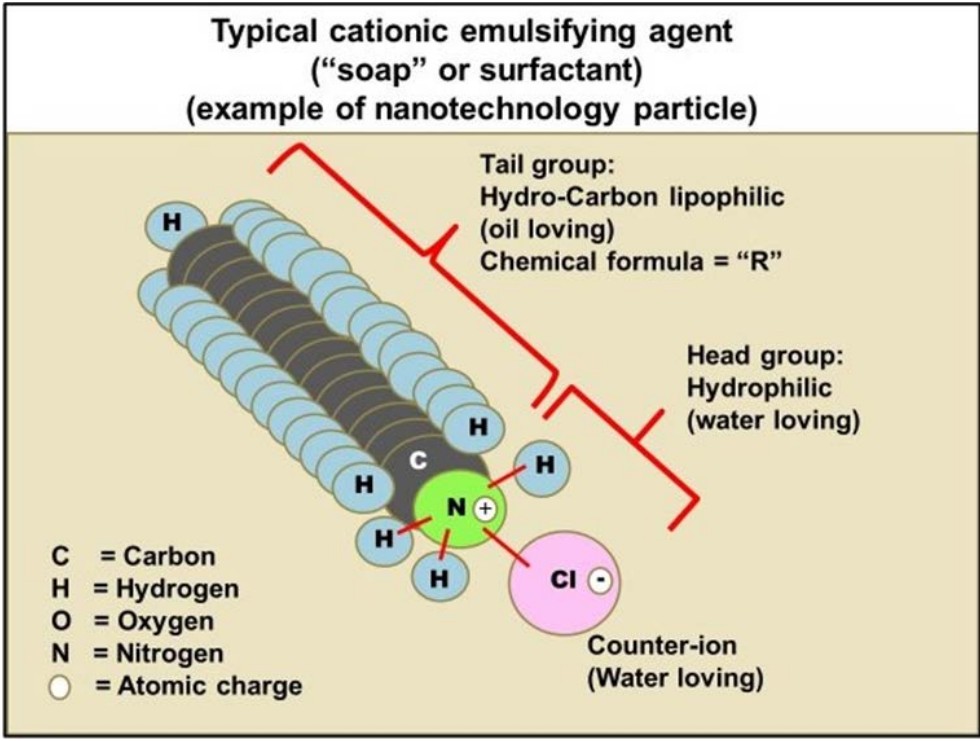

**Figure 2.** Typical composition of a cationic emulsifying agent.

The visual representation of the anionic emulsifying agent shown in Figure 2 is expressed in chemical formulation as follows [24,25]:

$CH_3(CH_2)_nCOO^- + Na^+$ (where n is normally a multiplier between 12 and 18 [17]:

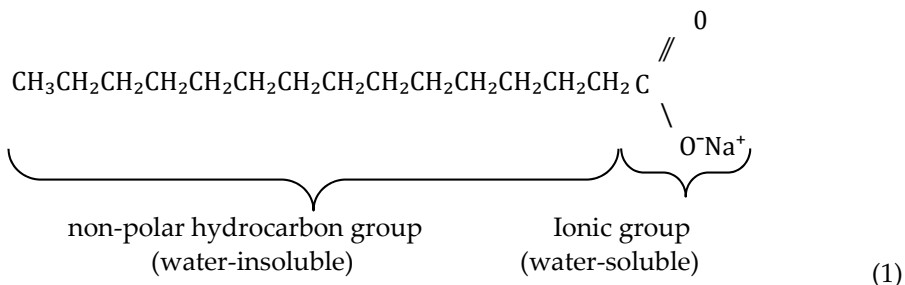

$$
\begin{array}{c}
\quad\quad\quad\quad\quad\quad\quad\quad\quad\quad\quad\quad\quad\quad\quad\quad O \\
\quad\quad\quad\quad\quad\quad\quad\quad\quad\quad\quad\quad\quad\quad\quad\quad \parallel \\
CH_3CH_2CH_2CH_2CH_2CH_2CH_2CH_2CH_2CH_2CH_2CH_2CH_2CH_2\ C \\
\quad\quad\quad\quad\quad\quad\quad\quad\quad\quad\quad\quad\quad\quad\quad\quad\quad\quad \backslash \\
\quad\quad\quad\quad\quad\quad\quad\quad\quad\quad\quad\quad\quad\quad\quad\quad\quad\quad\quad\quad O^-Na^+
\end{array}
$$

non-polar hydrocarbon group      Ionic group
(water-insoluble)            (water-soluble)      (1)

Equation (1) is simplified to:

$$
\begin{array}{c}
\quad\quad\quad O \\
\quad\quad\quad \parallel \\
CH_3(CH^2)_n\ C \\
\quad\quad\quad\quad \backslash \\
\quad\quad\quad\quad\quad O^-Na^+
\end{array}
$$

(2)

with the value of "n" typically varying between 12 and 18 [17].
Formula (2) is further simplified to:

$$
\begin{array}{c}
\quad\quad O \\
\quad\quad \parallel \\
"R"\ \ -\ \ C \\
\quad\quad\quad \backslash \\
\quad\quad\quad\quad O^-Na^+
\end{array}
$$

(3)

Similarly, a typical cationic emulsifying agent shown in Figure 3, is depicted as:

$$
\begin{array}{c}
\quad\quad\ H^+ \\
\quad\quad\ | \\
"R" -\ N^+\ -\ Cl \\
\quad\quad\ |\ \backslash \\
\quad\ H^+\ H^+
\end{array}
$$

(4)

The properties and stability of the emulsion is a function of numerous factors, including the chemical properties of the emulsifying agent (e.g., the length of the carbon-tail shown as "n"), the percentage of the emulsifying agent added during the emulsifying process, the manufacturing process and the properties of the bitumen. In terms of chemical stability, it is worth noting that the bond strengths between the various atoms in the emulsifying agent differ substantially. These bond strengths could also play a major role in the stability of the emulsion, especially in combination with a second nano-particle and/or when a modification to the emulsification agent is introduced. The bond strengths between some of the major atoms comprising the emulsifying agent are summarised in Figure 3 (compiled from published information [28]).

From Figure 3, it is seen that the bond strengths between the elements comprising an anionic emulsifying agent (pink arrow combinations) are considerably stronger than the bond strengths comprising the typical cationic emulsifying agent (green arrow combinations). This simplified chemistry explains the general trends found in the stability normally associated with anionic versus cationic bitumen emulsions in practice, assuming all manufacturing processes are optimised in line with good practices. The implication in practice is that an anionic nano-modified emulsion will normally have a longer shelf life (due to higher stability) than a nano-modified cationic emulsion, an important practical factor, especially considering remote areas of implementation, uncertain climatic conditions and a construction industry often faced with unplanned delays due to political and managerial

factors, not within the control of the contractor. The importance of the characteristics of the emulsifying agent nano-particle combined with a material compatible organofunctional nano-silane in the performance of the nano-modified emulsions as a stabilising agent for the treatment/stabilisation of granular materials is demonstrated in Section 7 of this article. The results show the variation in the engineering measured properties possible in practice with all input parameters carefully controlled. In this experiment, the only variable is the characteristics of the various emulsifying agents used by the various manufactures in the production of the nano-modified emulsions.

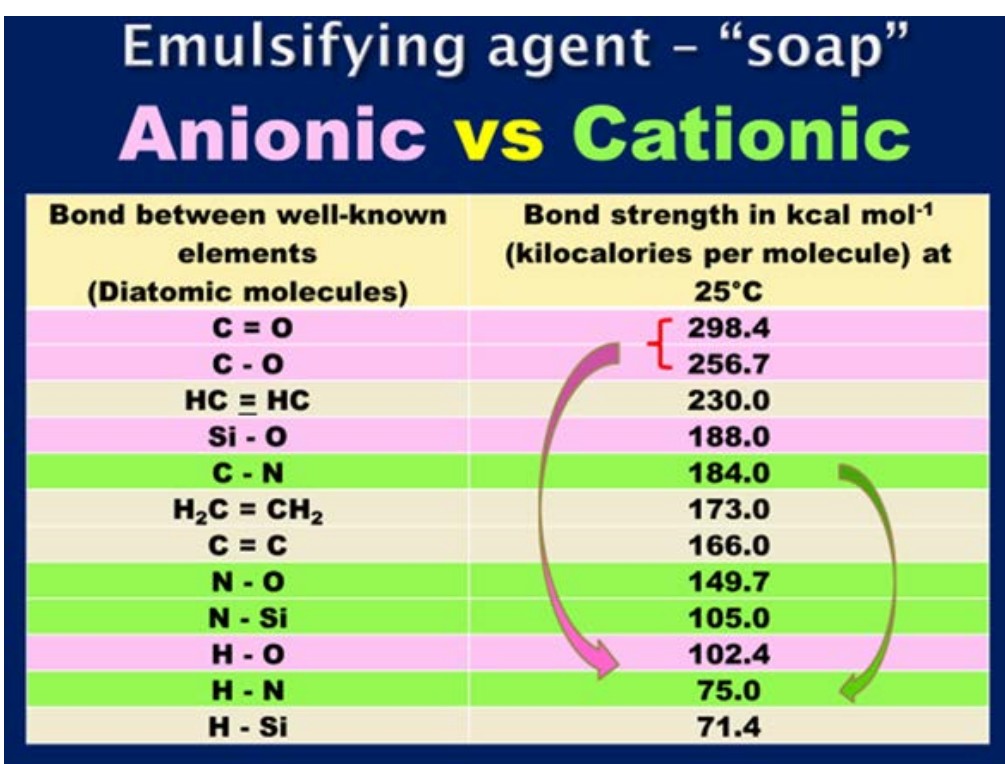

**Figure 3.** Comparison of the bond strengths between the various elements comprising the different emulsifying agents (anionic in pink and cationic in green) (compiled from published information [28]).

## 4. Basic Chemistry Applicable to the Understanding of the Successful Introduction of Material Compatible Nanotechnology Solutions for the Treatment, Improvement and/or Stabilisation of Granular Materials for Use in Road Pavement Layers below the Surfacing

### 4.1. Silicon Characteristics in the Material Sub-Strata and Applications within the Built Environment

Silicon (Si) is the second most abundant element (after oxygen), comprising more than 26 per cent of the crust of the earth (by weight). In comparison, oxygen makes up more than 49 per cent by weight of the crust of the earth. The rest of the elements contained in the crust of the earth in combination makes approximately 25 per cent by weight of the crust of the earth. Silicon is found in nature as oxides ($SiO_2$) or silicates ($SiO_4$), forming the basis of most rock-forming minerals [15]. Commonly known naturally available materials such as granite, feldspar, hornblende, asbestos, clay and mica are a few examples of materials normally containing high percentages of silicon [22].

Silicon is also one of the most useful elements to mankind. In the form of sand and clay, it is commercially used as a cost-effective product to produce building components such as pottery, bricks, cement, glass, etc. Silicon also plays an important role in plant as well as animal life, forming part of cell structures and found in plant remains as well as skeleton structures. It is considerably versatile in application, inherent to most of the

nanotechnology-based products currently in common use in the built environment, with "excellent mechanical, optical, thermal and electrical properties" [28].

The silicon present in nature as silicates can form 4 bonds with oxygen atoms, which may be orientated in various geometric structures, forming a three-dimensional infinite structure [29,30] to form minerals and the surface of the sub-strata or rock surface as commonly referred to. Figure 4 shows a very simplified illustration of the composition of naturally available materials (rocks), as numerous elements are found in nature that combines with the basic silicon lattice to form a large variety of minerals commonly found in all rock formations. Of importance is the illustrated attraction of exposed siliceous materials on the surface of granular materials to water molecules freely available in the atmosphere.

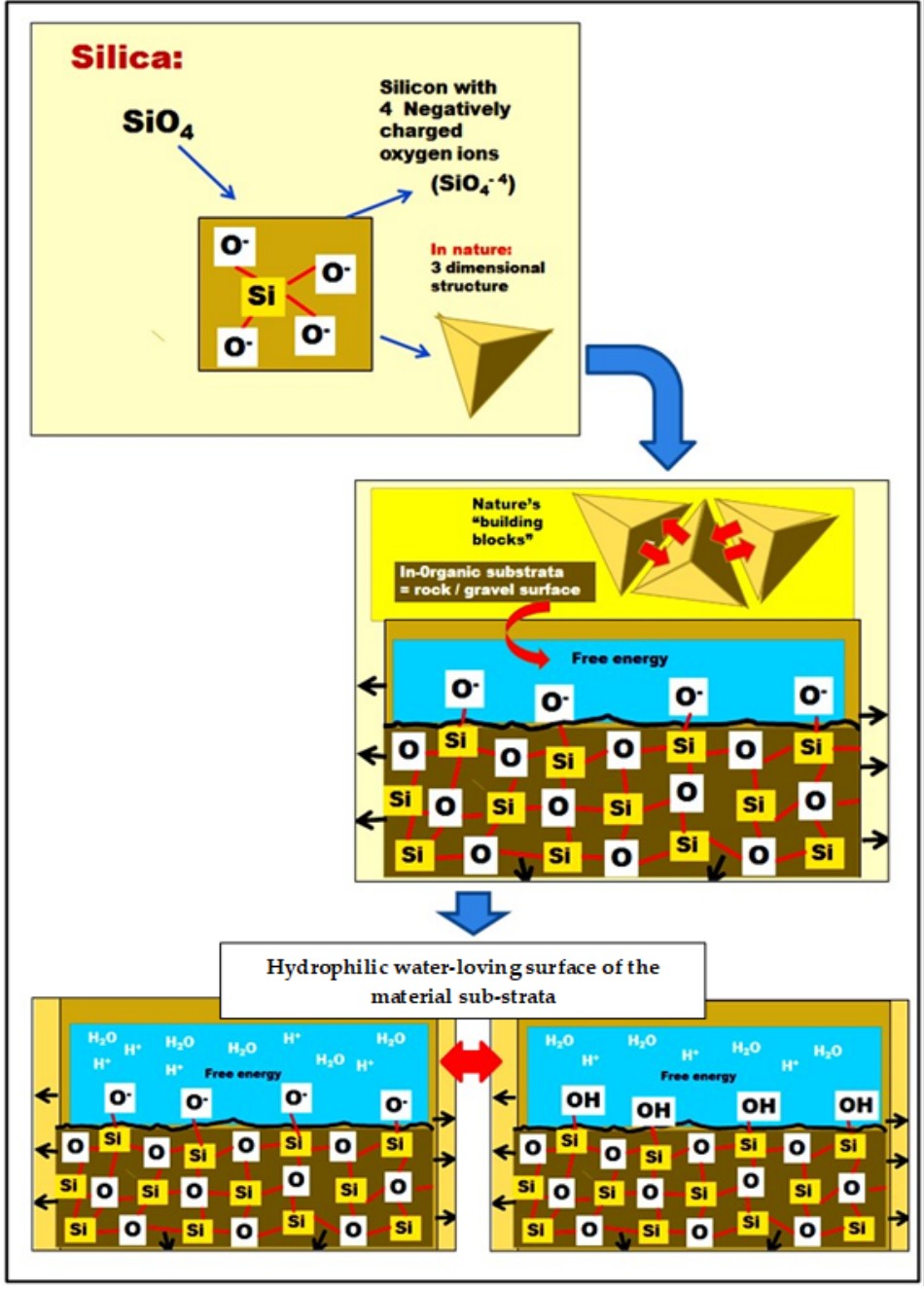

**Figure 4.** Simplified demonstration of the most common sub-strata or rock formations in nature.

Rock surfacings contain "broken chains" of chemical bonds that form a layer of free energy [15] on the surface of granular materials (rock/gravel or soil). The surfaces of

these naturally available materials (with a few exceptions, e.g., talk) are hydrophilic (water-loving) and attract water molecules freely available in the atmosphere. Hence, freshly crushed stone is instantaneously covered by numerous layers of water molecules that (a water molecule is approximately 0.1 nm in size or 1 Angstrom (Å)). These layers of water molecules covering the area of the stone are invisible to the eye and other senses (touch/smell). Water molecules ($H_2O$) are always accompanied by their natural derivatives of negatively charged oxygen ($O^{-2}$) and positively charged hydrogen ($H^{+1}$) in the atmosphere that surrounds the surface of granular materials (rock/gravel/soil) and is attached to the surface as shown in Figure 4.

The freely available hydrogen ($H^{+1}$) combines with the negatively broken oxygen chains of the Si elements to form $(OH)^{-1}$ bonds on the surface. The siliceous surfacings containing $O^{-2}$ and $(OH)^{-1}$ bonds are hydrophilic in nature and hence, attract water. The simplified lattice shown in Figure 4 forms the basic understanding of the application of various silane-nanotechnologies developed using the properties of the silicon element, which have been used in the built environment for more than 150 years to protect buildings, monuments and statues [4]. These same basic technologies are also directly applicable to the field of pavement engineering to enable generally available natural resources (granular materials) to be used more cost-efficiently within road pavement structures.

### 4.2. Chemistry of Organofunctional Nano-Silanes for Application in Pavement Engineering

Most of the nanotechnology products currently used in the built environment to protect, preserve and/or strengthen building materials are based on the silane ($SiH_4$) derivative of silicon (the same derivative, which has found application as the basis for the development of stone consolidants for more than a 150 years [4]). Each hydrogen (H) atom in the silane ($SiH_4$) can be replaced by any other element or group(s) to form a large variety of nano-products with numerous fields of application.

The most common reactive groups are hydrogen (H), chloride (Cl), fluorine (F) and a RO group, where R is the general symbol for the alkyl group, also known as the organo-functional group in silane-chemistry (also referred to as the hydrocarbon group discussed and explained in the composition of the emulsifying agents demonstrated in Figures 2 and 3), which may, inter alia, include chemical compounds such as $CH_3$ (methyl) or $CH_3CH_2$ (ethyl).

The $CH_3$-Si bond is a very stable (non-reactive) bond (known as organo-silane), with low surface energy with hydrophobic (water-repellent) or oleophilic (oil-loving) characteristics. The reactive group RO is referred to as an alkoxy (alkyl + oxygen) group [4]. Silicon (Si) forms the centre of the RO group together with the second functional group(s) (X) (e.g., methoxy, ethoxy, etc.) to form the chemical molecule RO-Si-X. These secondary functional (pendant) groups may be highly reactive (in cases where the element of carbon (C) is excluded from the formulation) and define the functionality of a specific nano-silane [4]. The functional group (R) will attach to organic material (such as bitumen or an alternative material compatible polymer with an oil basis), while the functional group(s) (X) will attach to inorganic material (granular material such as rock/gravel/soil). Hence, an organo-functional nano-silane combines the functionality of a reactive group with a non-reactive group in a single molecule.

It should be noted that the organofunctional nano-silane is not replacing the stabilising agent in the stabilisation of granular road building materials. The molecule is too small to effectively bridge the gaps between the material fractions. However, in practice, the nano-silane can very effectively fulfil the role of a bonding agent (adhesive agent, a well-known concept in the asphalt industry), which is designed to permanently bind the stabilising agent (the bitumen or equivalent organic substitute with similar characteristics) to the granular materials (stone/aggregate/soil). Si–O chemical bonds are some of the strongest found in nature (refer to Figure 3). The relatively small size of the nano-silane particle (depending on the type of nano-silane and the quality of the product, the size of nano-silanes may vary from about 5 nm to less than 1 nm) results in a considerably high surface

area per volume ratio [7]. In other words, a relatively small amount of the nano-silane is required to totally encapsulate all particles of the granular material (stone/gravel/soil) that is being treated/stabilised (for example, 1 litre of nano-silane could easily have the same coverage area of about a 1000 litres of bitumen, with a bitumen particle being in the order of 1000 to 5000 nm in size) [7,17]).

As mentioned, the organofunctional nano-silane particle will, in effect, have the properties of a "coupling agent" commonly referred to in the asphalt industry as an aggregate adhesive. Aggregate adhesives or aggregate promoters are well-known terminologies in pavement engineering, and such nanotechnologies have been used for at least three decades [31] as modifiers to bitumen emulsions. However, the use of silicon-based nanotechnology products in pavement engineering is relatively new and has been developed mainly over the last 15 years.

The functionality of the silane R-Si-X molecule in terms of an aggregate adhesive mainly refers to the ability of the (X) reactive group to be removed from the molecule when in contact with water (to effectively be chemically detached by water) and replaced by a hydroxyl (OH) group, during a process referred to as hydrolysis [4]. where:

$$\text{Hydrolysis: Si-X} + H_2O = \text{Si-OH} + XH \tag{5}$$

The one product of hydrolysis is the Si-OH binding referred to as silanol [4]. Silanols are now able to react with each other to form siloxane bonds in a condensation reaction when in contact with an inorganic material (e.g., granular material such as rock/gravel/soil) [4], covering the total area of each particle in the granular material. It should be noted that the by-product during condensation should preferably be water ($H_2O$), as shown in the example (not all available nano-silane products produce water as a by-product, at worst a non-toxic alcohol should be allowed to form as a by-product during condensation—refer toxicology requirements [7]). where:

$$\text{Condensation: Si-OH} + \text{OH-Si} = \text{Si-O-Si} + H_2O \tag{6}$$

Of importance to the field of pavement engineering is that the organic (R) (organofunctional) group attached to the silicon by means of a direct Si-C bond is not affected by hydrolysis and will remain stable (non-reactive) also during condensation. It follows that in the case of $CH_3\text{-Si-}(OCH_3)_3$ (methyl-trimethoxy-silane), the methyl group ($CH_3$) directly attached to the silicon atom will remain stable throughout the process of hydrolysis and condensation, while the hydrogens in the methoxy (($OCH_3)_3$) groups will react with water. These effects of hydrolysis and condensation are demonstrated as follows [4]: where:

$$\text{Hydrolysis: } CH_3\text{-Si-}(OCH_3)_3 + 3H_2O = CH_3\text{-Si-}(OH)_3 + 3CH_3\text{-OH} \tag{7}$$

Condensation:

$$CH_3-Si-(OH)_3 + (OH)_3-Si-CH_3 = CH_3-\overset{\overset{\displaystyle OH^-}{\displaystyle |}}{\underset{\underset{\displaystyle OH^-}{\displaystyle |}}{Si}}-O-\overset{\overset{\displaystyle OH^-}{\displaystyle |}}{\underset{\underset{\displaystyle OH^-}{\displaystyle |}}{Si}}-CH_3 + H_2O \tag{8}$$

In order to be effective as a consolidant, the silane-compounds must be able to form three-dimensional networks and hence, must have at least three reactive groups. The pavement engineering implications (macro-effect) of the above nano-silane formulations are demonstrated in Figure 5, which should be considered together with Figure 4 and the similarities with the emulsifying agent shown in Figures 1 and 2.

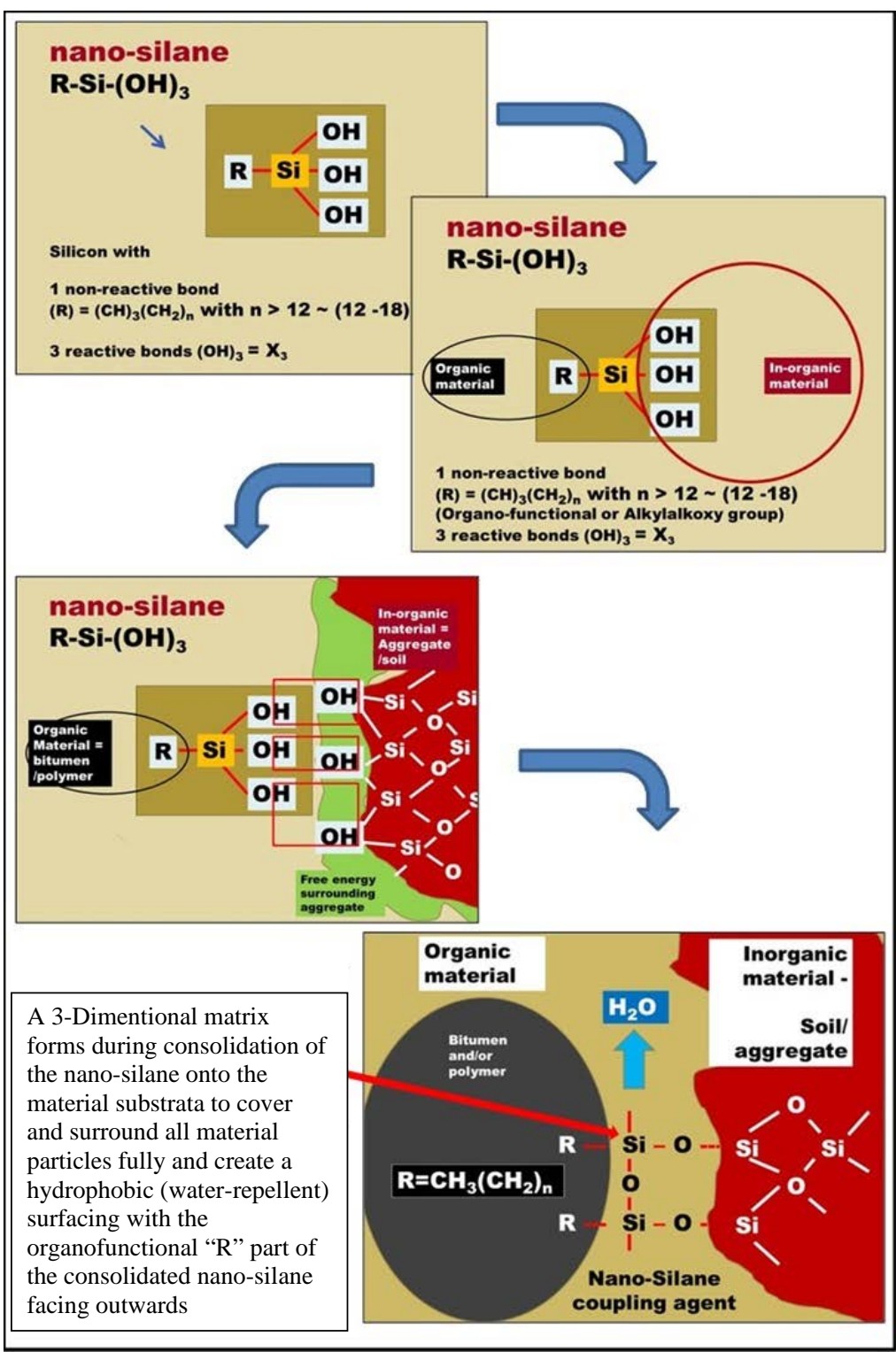

**Figure 5.** Simplified demonstration of the chemical interaction between a stabilising agent, organofunctional nano-silane modification and the mineralogy of the rock/stone/soil granular material sub-strata.

The effective result shown in Figure 5 is that the surface of each particle comprising the granular material (rock/aggregate /soil) will chemically be changed from a hydrophilic (water-loving) to a hydrophobic (water-repellent) state that attracts an organic material (such as oil/bitumen or an equivalent material compatible polymer). Consequently, the

water molecules that are naturally attracted to exposed surfaces of granular materials (stone/aggregate/soil) due to the presence of broken chemical bonds as discussed and demonstrated in Figure 4, will now actively be repelled together with the water that is the by-product formed during the condensation phase (as per the example shown in Figure 5).

Figure 5 represents a most basic schematic explanation of the practical use of nano-silane science in pavement engineering. The changing and matching of reactive and non-reactive bonds to the silicon element can result in literary numerous different nano-products with different characteristics, which could match the numerous minerals available in naturally available materials such as stone, gravel or soils.

At the same time, through creating the appropriate bonds, the surface of these materials will become water repellent, negating the negative impact that water has on the durability of materials through the prevention of (or at least limiting) weathering due to chemical decomposition (a pre-requisite of the process of weathering of materials through chemical decomposition (chemical change) and the formation of resultant secondary minerals in the presence of and access to water [15]). In pavement engineering design analyses, it is usually assumed that material mineral properties will stay unchanged during the design period. This basic assumption is incorrect, and even over a design period of 20 to 40 years, dramatic changes can occur (even in freshly crushed stone—given conducive environmental conditions [15]), which will influence the design assumptions considerably.

This aspect alone shows the potential benefit of the use of organofunctional nano-silane products in pavement engineering as a protective agent for high-value, freshly crushed stone against chemical decomposition and deterioration over its design life (e.g., a 20-year normal design period). Rehabilitation investigations conducted on major freeways in the Gauteng province of South Africa have shown that high-quality newly constructed stone G1 (Figure 6) [32] base-course material can deteriorate over a period of 22 years to an equivalent G4 to G6 quality material. Similarly, secondary roads in the Gauteng region of South Africa have shown that a cement-treated base-course layer can deteriorate from a cement-treated layer (Unconfined Compressive Strength (UCS) between 750 to 1500 kPa) quality layer to an equivalent G7/8 quality material [32], over a period of 20 years (rehabilitation investigations undertaken in the Gauteng Province of South Africa by the main author).

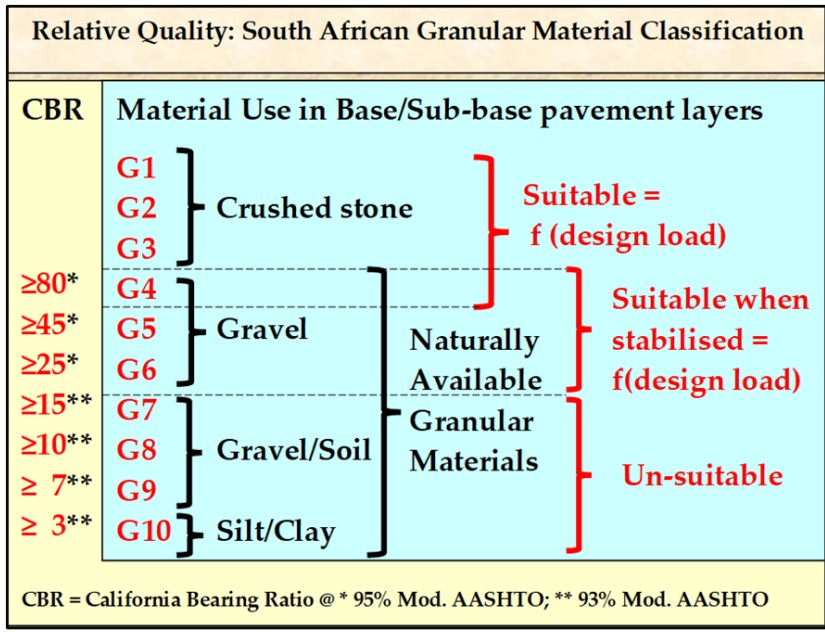

**Figure 6.** Classification of granular materials (comparative summarisation of some characteristics of the material classification [32]) (refer CBR compaction [33,34]).

### 4.3. Modification of Nano-Silanes through Hydroxy Conversion Treatments (HCT) Based on the Mineralogy of the Granular Material Sub-Strata

Although approximately 60 percent of the crust of the earth contains Si-O bonds, not all naturally available granular materials are silicon-rich. Non-silicate materials include carbonate materials such as calcretes and dolerites that are predominant in certain areas of the world and laterites (abundant in warm/humid conditions conducive to chemical weathering and the leaching-out of silicate materials). In such cases (e.g., with calcretes, dolomites, etc.), very little or even no silicon may be available in the material to create the Si-O-Si bonds, as explained in the example.

The part of the earth conducive to weathering due to chemical decomposition is situated in areas associated with the presentence of moisture (humid conditions) and high variations in seasonal temperature variations. These environmental conditions are especially pronounced between the Tropics of Capricorn and Cancer. The mineralogy of the naturally available materials (especially in areas close to the Equator), may be deprived of readily available silicon and may not be able to form Si-O bonds with the nano-silanes, as shown in Figure 5. In these areas, the nano-silanes need to be complemented with a material compatible nano-particle conversion to created strong chemical bonds with the minerals in the available granular materials. This conversion of a nano-silane is generally referred to as a Hydroxy Conversion Treatment (HCT), which is best characterised by its ability to form bonds with the available minerals in the granular materials (stone/gravel/soil). For example, in the case of carbonate-rich materials (e.g., dolomites and calcretes), the nano-particle added to the nano-silane in the form of an HCT must be able to form strong chemical bonds with the Calcium-carbonates ($Ca(CO_3)$) in the material. The example of an HCT treatment of a nano-silane for use with carbonate-rich material is demonstrated in Figure 7, where the added HTC is designed to form strong chemical bonds with the carbon elements in the material sub-strata.

From the above example, it should be clear that the different nano-products applicable to various mineral compositions must accurately be identified through scientific testing [20] to obtain the required information to identify a material compatible nano-silane product, with or without the addition of an applicable HCT, in order to achieve the required results to eliminate the risk of failure. Similar conversions will be required for other materials that may contain minerals with a relatively low free silicon content. For example, laterite may contain anything from 90 percent to 10 percent silicon-based minerals [15], which may, inter alia, depend on the parent material (primary minerals), the climatic conditions, and the extent of chemical weathering that has taken place [22].

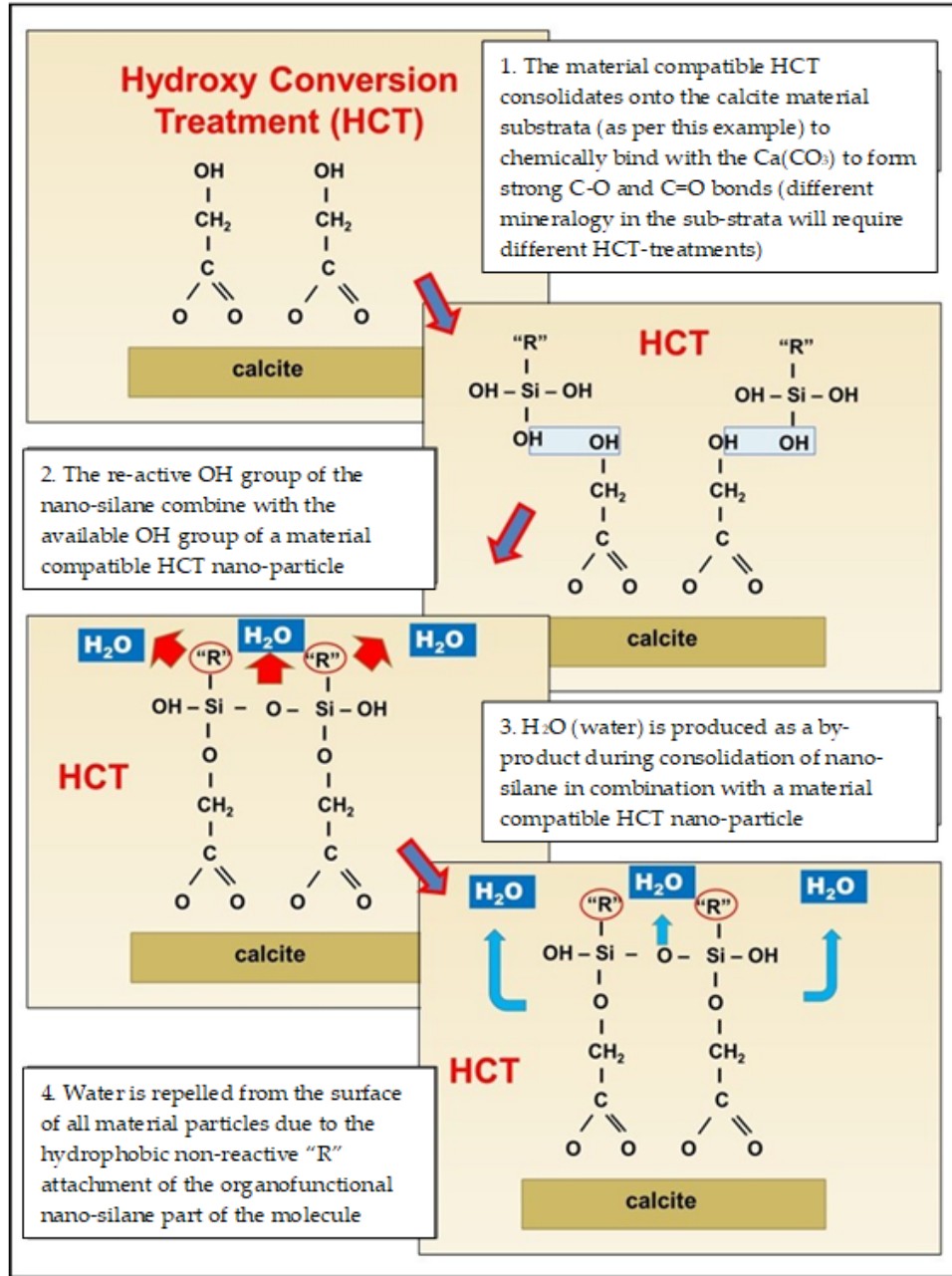

**Figure 7.** Hydroxy Conversion Treatment (HCT) of a nano-silane modification to a stabilising agent to enable a strong bond with carbonate materials.

## 5. The Clay Problem

The identification of a material compatible HCT treatment in the case of highly chemically weathered materials may be more complex due to the high percentages of secondary minerals with little silicon present in the naturally available granular materials. The predominant minerals present in these materials are usually closely associated with a high percentage of clay within the materials. Such materials may present additional challenges in terms of practical constraints associated with workability. Due to the sub-nano size of some clay particles (Figure 8 [35]) and the formation of nano-sheets by high percentages of clays, the nano-silane particle size to be added needs to be able to penetrate the clay matrix in order to achieve a hydrophobic stable state within the material. In the absence of such internal hydrophobicity, the clay sheets will shear under loading, breaking any protective stabilising agent and allow water to access the clay crystals within the clay

matrix, rendering the hydrophobicity of the stabilised material to become ineffective. The stabilising agent also needs to be of nano-size in order to effectively bind the material together in combination with the applicable nano-silane.

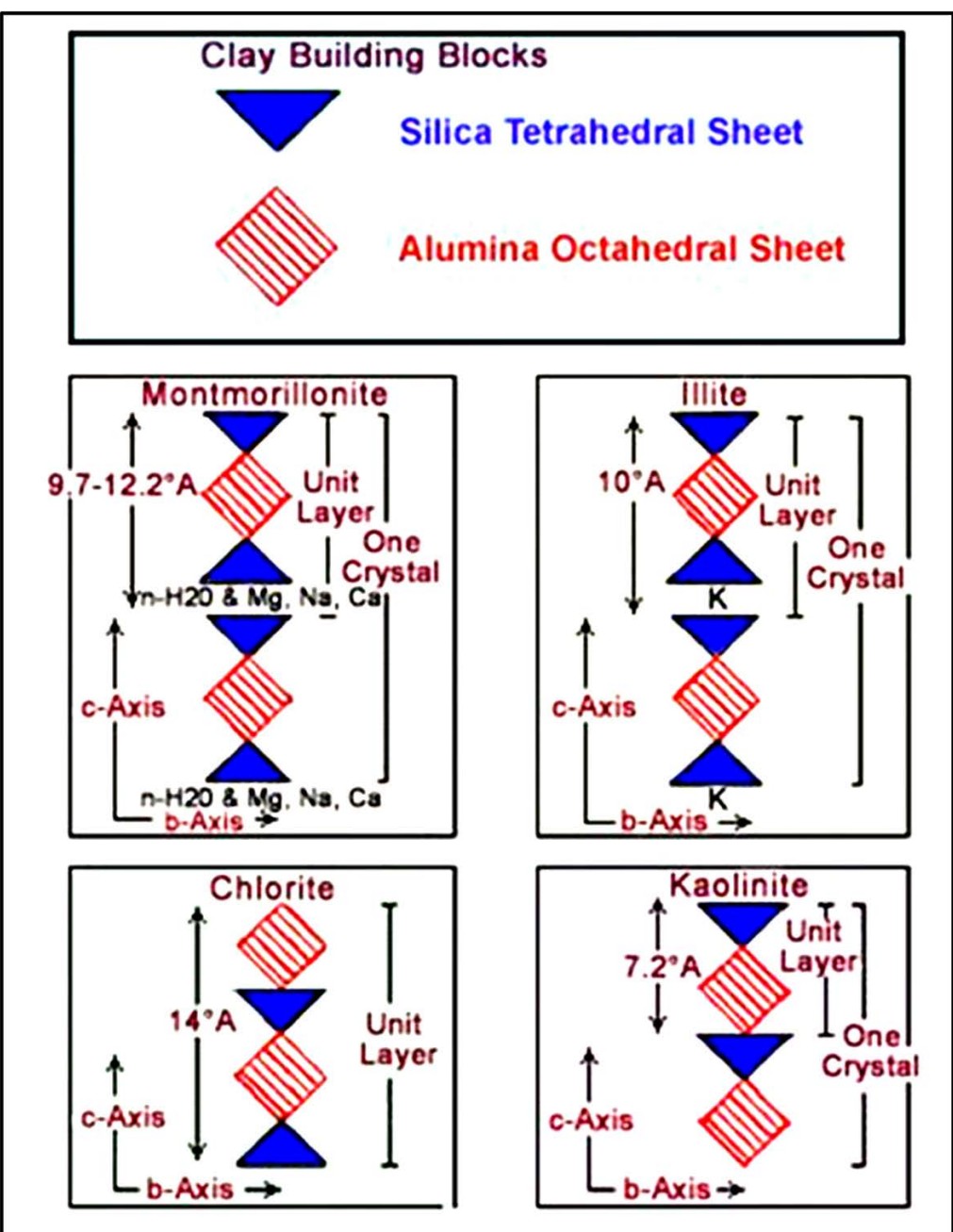

**Figure 8.** Typical dimensions in Angstrom of commonly available clay crystals that form during chemical decomposition within naturally available materials [35].

A micro-size molecule in the stabilising agent will prove problematic with the nano-size granular material particles (clay particles) effectively "swimming" within the relatively large particles of the stabilising agent with no bonding between the material particles possible. This aspect is practically demonstrated by the material test results shown in Figure 9. The material stabilised with a material compatible New-age (Nano) Modified Emulsion (NME) stabilising agent, as shown in Figure 9, contained between 20 and 25 percent of fines passing through the 0.075 mm sieve of which more than 60 percent was identified as mica, clay and talk minerals. The NME stabilising agent is an anionic NME manufac-

tured using a bitumen emulsion. It is seen that an increase in the stabilising agent from 1 percent to 1.5 percent led to a decrease in the Unconfined Compressive Strength (UCS) in a dry condition just after rapid curing [18] (UCS$_{dry}$) from about 4800 kPa to 2600 kPa (highlighted in pink), while the Indirect Tensile Strength (ITS) in a dry condition after rapid curing [18] (ITS$_{dry}$) decreased from 420 kPa to 230 kPa (highlighted in pink). Although these results are still very good considering the quality of the material that was stabilised (G8—refer comparison in Figure 6) (demonstrating the considerable advantages of the use of a material compatible anionic NME stabilising agent), the decrease in the measured engineering properties are considerable. This decrease can directly be attributed to the relatively large particle size of the stabilising agent (1000 to 5000 nm) together with the relatively high percentage of the material passing through the 0.075 mm sieve size of which a large percentage can be considered as problematic materials (i.e., clay, mica and talk).

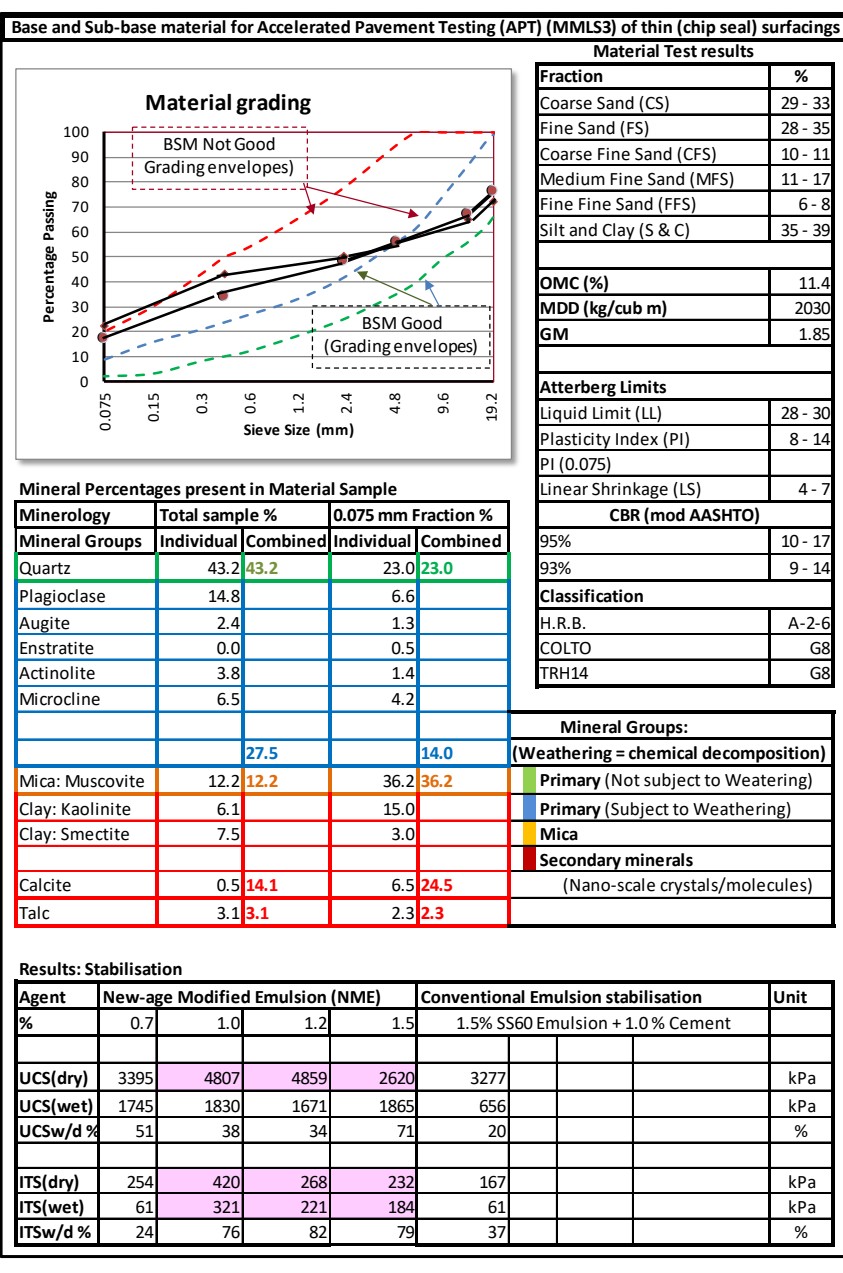

**Figure 9.** Results from the anionic NME stabilisation (using a bitumen emulsion) of a material containing between 22 and 25 per cent of fines passing through the 0.075 mm sieve size of which more than 60 percent consists of mica, clay and talk particles.

## 6. Material Compatible Anionic NME Nanotechnology Stabilising Agents Enabling the Use of Unsuitable Granular Materials to Become a Valuable Asset in Road Construction

The example shown in Figure 9 is a good indicator of the disruptive nature of the introduction of proven available nanotechnologies into the field of pavement engineering. Using traditional material classification systems, the indicator material test results shown in Figure 9 would have identified the material to be unsuitable for use in any load-bearing layers of a pavement structure, even for low-volume roads. However, the stabilisation results gave confidence to the use of the material in the sub-base with 1 percent anionic NME stabilising agent (highest results in terms of tensile strength) and with 1.5 percent in the base layer (highest Retained Compressive Strength (RCS) of 71 percent).

A high RCS can be related to high resistance to possible damage due to water-ingress from the top in the case of a compromised surfacing layer. This pavement structure constructed with the anionic NME stabilisation as indicated was evaluated using Accelerated Pavement Tests (APT) [8,9]. The design traffic loading of 3 Million Standard 80 kN dual-wheel single axle loads (E80s) was surpassed with ease with an estimated load-bearing capacity of at least three times that of the design traffic loading in terms of applicable pavement deformation performance criteria [36]. The stabilised materials have also shown remarkable resilience against overloading with a damage co-efficient of n < 2 (normally assumed to be 4.2 [37]). It follows that road pavement structures containing anionic NME stabilised base and sub-base layers will be ideally suited for the construction of roads where law enforcement in terms of vehicle over-loading is limited or even totally absent.

Available nanotechnologies have been used under laboratory conditions to achieve acceptable engineering results on materials of considerably poorer quality than shown in Figure 9. Figure 10 shows the exposed surface of a material sample crushed during an $ITS_{wet}$ test of relatively poor laterite. This material contained almost 50 percent of fines passing through the 0.075 mm sieve size, of which almost 70 percent was identified through XRD scans as kaolinite clay particles. In this case, a material compatible HCT modified high-quality sub-nano size silane was used with a nano-polymer to stabilise the material. The crushed sample shown in Figure 10 was tested to have an $ITS_{wet}$ of 80 kN with good hydrophobicity achieved throughout the sample as shown by the beading [7] effect of the water sprayed onto the exposed surface of the broken sample after testing. These anionic NME treated materials will be suitable for use in the upper load-bearing layers of some low-volume roads (refer criteria [38]). The sub-nano size silane and nano-polymer needed to be carried into the material using water as a carrier and thoroughly mixed. As well known, high percentages of clay in a wet condition present considerable challenges in terms of workability in the field. In practice, workability may need to be addressed by mixing a percentage of material such as sand or naturally available gravel into the material to practically enable the material to be workable when adding the modified stabilising agent into the material using water as a carrier.

The material shown in Figure 10 may be considered to be an extreme example of a granular material not to be considered for use in traditionally accepted pavement engineering. However, such materials are all too common in areas close to the equator, associated with high humidity and high-temperature conditions. Granular materials of similar characteristics (shown in Figure 10) may be the only material available over considerable distances. Hence, any improved use of granular materials of very poor quality, similar to that demonstrated in Figure 10, may be of considerable cost benefit to any road project [39].

The use of materials of such a poor quality may be unthinkable in traditional pavement engineering. However, this extreme example demonstrates that with scientific knowledge about the mineralogy [20], in combination with applicable basic knowledge of the basic chemistry, material compatible nano-silane technologies, modified with a material compatible HCT, can be selected and utilised to change the material characteristics to an extent where even poor quality laterites can effectively be treated to become hydrophobic and

stabilised to an extent where it can meet the engineering specifications of load-bearing layers in a pavement structure [35].

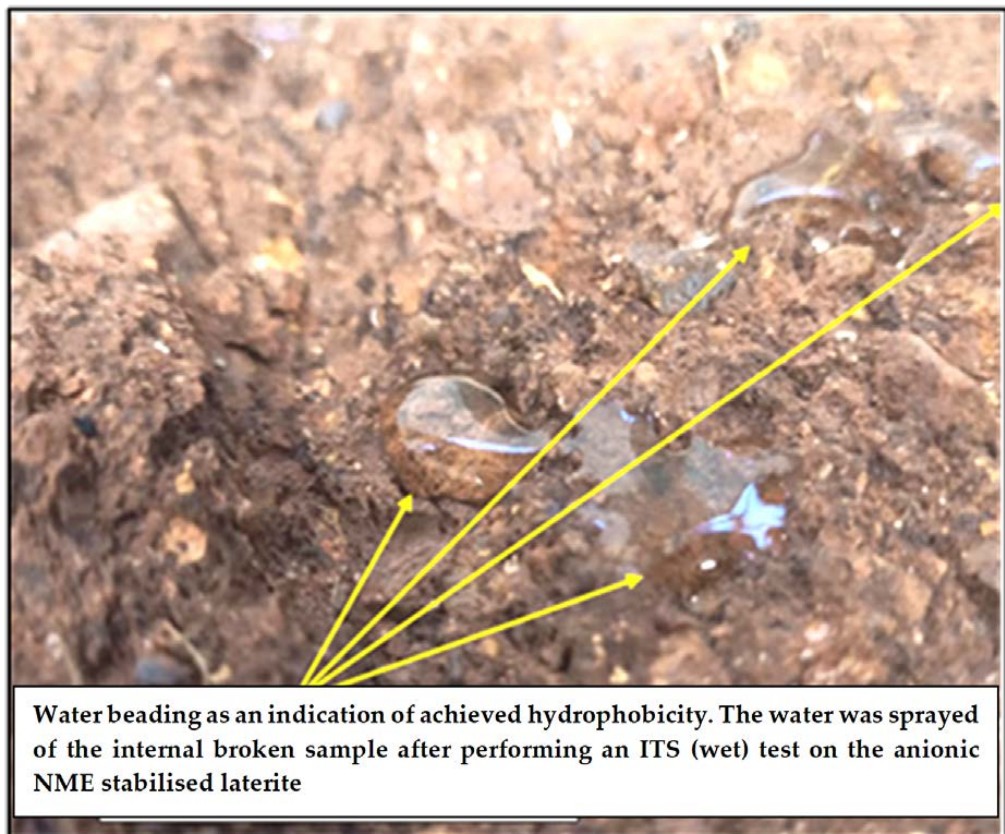

Water beading as an indication of achieved hydrophobicity. The water was sprayed of the internal broken sample after performing an ITS (wet) test on the anionic NME stabilised laterite

**Figure 10.** Water beading of water spayed on an exposed laterite material sample stabilised with a material compatible HCT modified sub-nano size anionic silane and graded nano-polymer after crushing while performing an $ITS_{wet}$ test—almost 50 percent of the material passed through the 0.075 mm sieve size of which almost 70 percent was measured to be Kaolinite clay particles.

## 7. Influence of the Chemical Characteristics of the Emulsifying Agent on the Potential Engineering Properties Achievable

A good emulsifying agent (i.e., an emulsifying agent with a relatively long carbon tail) also contributes significantly towards the engineering properties to be achieved when used for the stabilisation of granular materials for use in roads. These characteristics are normally not addressed in specifications (e.g., [23,40–42]). It follows that bitumen emulsions manufactured according to such specifications could result in substantial differences in the engineering properties when used to stabilise the same materials [43]. Under conditions where procurement procedures are based on the award of tenders based mainly on the lowest tender, modified bitumen emulsion manufacturers would need to produce the modified bitumen emulsions within specification using the lowest possible input costs. The lowest cost emulsifying agents are normally also associated with carbon tails (Figures 1 and 2) of limited length.

The considerable influence of the chemical characteristics of the emulsifying nanoparticle (surfactant) as part of an anionic NME stabilising agent is demonstrated through the use of eight different anionic NME stabilising agents for the stabilisation of the same material. These eight different anionic bitumen emulsions were sourced from five different suppliers (some using more than one different emulsifying agent). Variations in bitumen characteristics were limited by ensuring that all suppliers use bitumen produced by the same refinery within the same time frame. External factors other than the type of emulsifying agent are further reduced by modifying the different emulsions using the exact same

type and volume of material compatible nano-silane from the same flow-bin. The resultant anionic NME stabilising agents were used on the same material sourced at the same time in the same quarry, thoroughly mixed to ensure that material variations will have a limited impact on results. The granular material used was a relatively good G5/6 (refer to Figure 6) granite containing about 10 percent Muscovite. The material was stabilised using 1 percent per mass (containing about 0.5 percent residual bitumen) of the anionic NME (about 21 litres/$m^3$ of granular material) to stabilise the granular material. All samples were mixed, cured and tested in the same laboratory using the same technicians in a blind testing regime. The UCS tests (dry and wet [18]) and the ITS tests (dry and wet [18]) of three samples each were measured and compared using the eight different bitumen emulsions, all manufactured meeting the applicable National Standards [23].

The results using the eight different nano-modified bitumen emulsions manufactured using different emulsifying agents are shown in Figure 11 (UCS dry and wet), Figure 12 (ITS dry and wet) and Figure 13 (Retained Compressive Strength (RCS) ($UCS_{wet}/UCS_{dry}$ as a percentage) and Retained Tensile Strength (RTS) ($ITS_{wet}/ITS_{dry}$ as a percentage). The different suppliers are identified by the letters A to E. The number of emulsions prepared using different emulsifying agents received from each supplier is shown by the number following the alphabet letter, e.g., A1, A2. Where the emulsifying agent is known and to compare the same emulsifying agents from different suppliers, these emulsifying agents are identified by a further number, e. g, A1-1 and D1-1.

Although all the bitumen emulsions used conformed to the National standard [23] currently used in practice in South Africa, some alarming variations in the measured engineering properties of UCS, ITS, RCS and RTS were measured. Depending on the property tested, a variation of 30 to more than 50 percent in the average measurements between the eight different anionic NME products were recorded. Given the relatively good material (CBR > 45 @ 95% mod AASHTO [32,33]) used to compare the engineering properties, these results are alarming. Of particular concern is the comparison of the retained strengths. It is seen that the effect of the nano-silane modification (specifically added to improve granular materials against the effect of water) could be affected to a concerning degree by the chemical characteristics of the emulsifying agent (surfactant). It follows that both the nano-particles (emulsifying agent) used to manufacture the emulsion and the material compatible nano-silane are of importance to achieve optimum results. The use of a relatively inferior emulsifying agent in the manufacturing of the bitumen emulsion could result in the manufacturing of an anionic NME stabilising agent not meeting the engineering specifications, especially when used in combination with naturally available granular materials of relatively poor quality.

It follows that the manufacturing of bitumen emulsion not specifying the characteristics of the emulsifying agent can be problematic in practice and can lead to unforeseen construction problems. In an open procurement process, the achieving of the engineering properties is the prime objective. Hence, the use of inferior products can be limited through the implementation of an end-product specification. In such cases, the contractor and supplier must guarantee that the product to be used will be of a quality suitable to meet the minimum engineering properties as specified. Such an approach will ensure that the end-product will meet the engineering requirements of the design and limit any risk associated with chemical variations. During the design process, the design engineer must do sufficient tests to ensure that an available NME stabilising agent can meet the specifications with confidence. As a result of these findings, it is recommended that a high-quality modified emulsifying agent (modification of a high-quality emulsifying agent with a material compatible nano-silane) be used for the manufacturing of the NME stabilising agent to confidently meet the engineering specifications and limiting any risks associated with the possible use of an inferior component.

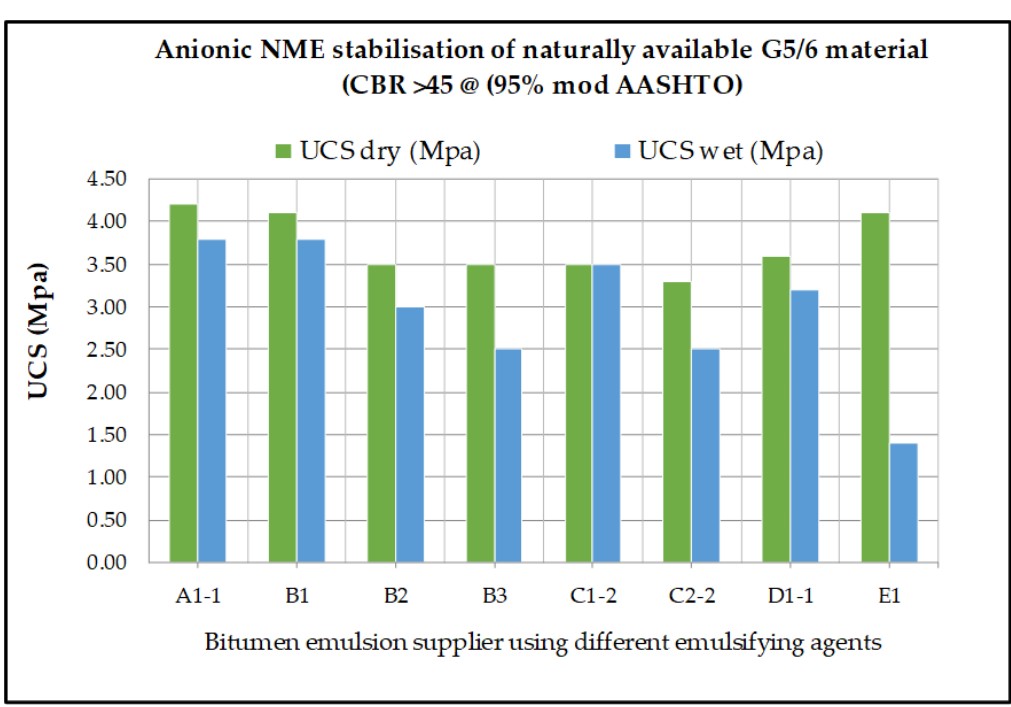

**Figure 11.** Comparison of the UCS results (dry and wet) using eight different bitumen emulsions from five different suppliers, with variations in the emulsifying agent (surfactant) used in the manufacturing of the anionic NME as the only variable– all anionic bitumen emulsions used were produced meeting the National Standard [23].

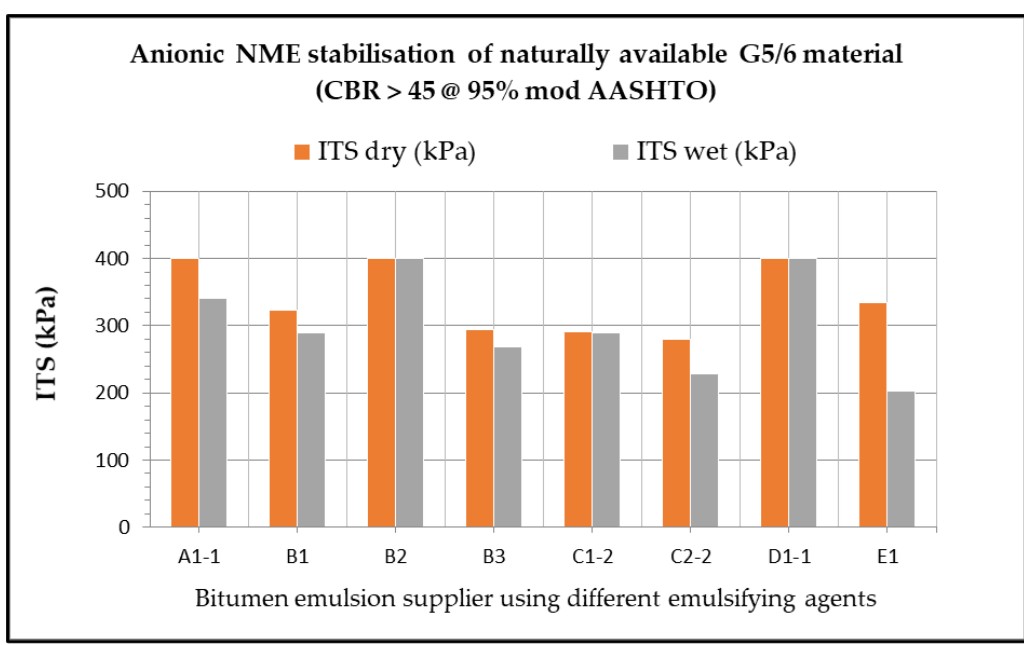

**Figure 12.** Comparison of the ITS results (dry and wet) using eight different bitumen emulsions from five different suppliers, with variations in the emulsifying agent (surfactant) used in the manufacturing of the anionic NME as the only variable—all anionic bitumen emulsions used were produced meeting the National Standard [23].

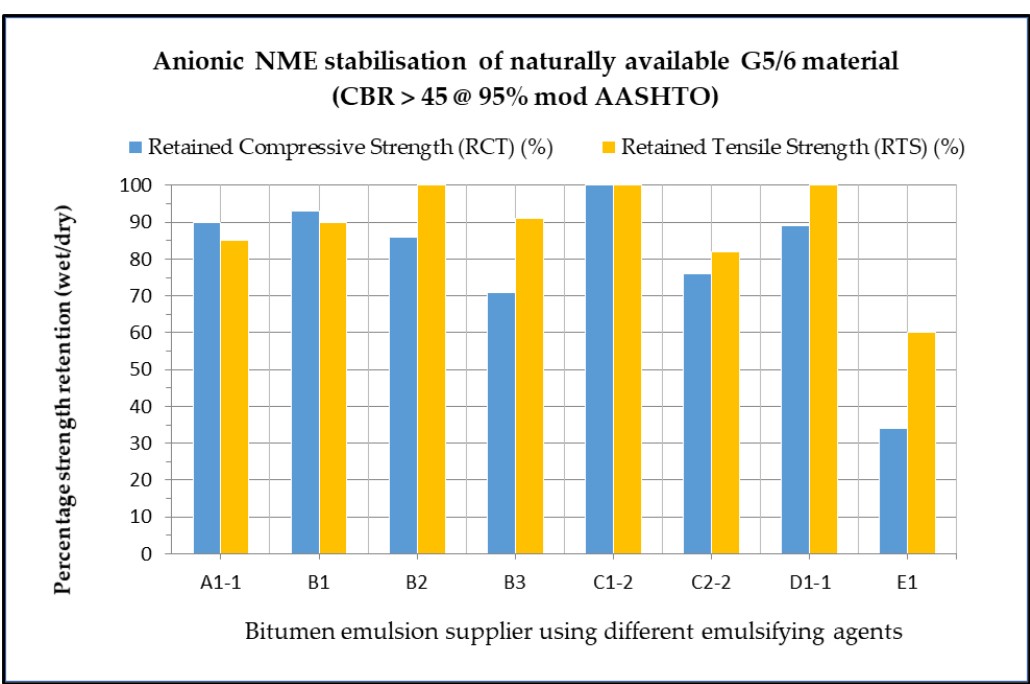

**Figure 13.** Comparison of the relative retained UCS strengths (RCT) ($UCS_{wet}/UCS_{dry}$ as a percentage) and relatively retained ITS strengths (RTS) ($ITS_{wet}/ITS_{dry}$ as a percentage) using eight different bitumen emulsions from five different suppliers, with variations in the emulsifying agent (surfactant) used in the manufacturing of the anionic NME as the only variable—all anionic bitumen emulsions used were produced meeting the National Standard [23].

## 8. Conclusions

Nano-silane based products have been used in Europe for almost two centuries to protect stone buildings against the impact of climate and pollution. Early in the 1800s, the development of these products was performed on a trial-and-error basis, with scientists often encountering conflicting results. These conflicting results soon led to the realisation that the "type of stone" (primary minerals) and the "condition of the stone" (secondary minerals) are important aspects that determine the successful outcome of the application of a specific nano-scale silicon-based product developed for the protection/stabilisation of granular materials (stone/gravel/soil).

Over the last few decades, numerous silicon-based nanotechnology products have been developed for use in the built environment, utilising the "excellent mechanical, optical, thermal and electrical properties" inherent in the silicon element. The same nanotechnologies, incorporating the "lessons learnt" from the built environment, can find direct application in road pavement engineering to enhance, protect and stabilise naturally available granular materials that are traditionally considered marginal or inadequate for use in the upper pavement layers below the surfacing.

In order to limit risks and eliminate the possibility of failure, engineers need to take cognisance of the lessons learned in the built environment. It is essential to know the scientifically determined mineralogy of the material that is available for use. In addition, it is also of importance to understand the basic chemistry involved in the use of various nanotechnology products. This understanding is instrumental to recognising the advantages and limitations of any one available material compatible product in combination with the naturally available granular materials for use in a pavement structure. Basic knowledge of elementary chemistry is considered essential to ensure that applicable technologies are used and that the impact on engineering results is understood and any potential risk is identified and timeously addressed.

Unmodified bitumen emulsions do not create chemical bonds with the material substrata that when used as a stabilising agent. Bond strengths are achieved through electrical

and mechanical forces and absorption into the material surfacing cervices. However, strong chemical bonds can be achieved through the introduction of material compatible nanotechnology organofunctional nano-silane modifications to the bitumen emulsion (or any equivalent polymer stabilising agent). The chemical interaction of the nano-silane modifier can be compared to the use of an emulsifying agent (nanoparticle) to create bitumen emulsions, which enables water to be mixed with oil (bitumen). The characteristics and the bond-strengths of the elements comprising a specific emulsifying agent will also affect the stability of the emulsion and the properties of the end-product. Changes in chemical composition of the emulsifying agent (also known as a surfactant or "soap") can meaningfully influence the engineering properties that can be achieved using a modified bitumen emulsion with the addition of a material compatible nano-silane.

The influence of the chemical characteristics of the emulsifying agent as the only variable in an anionic NME mix is demonstrated using eight different products from five different suppliers. Differences of up to 50% in the measured UCS (dry and wet) and ITS (dry and wet)) were measured using different emulsifying agents as the only component variable. It is recommended that this aspect be addressed in practice by engineers through the introduction of end-product specifications. The introduction of end-product specifications will put the onus (and risk) on the contractor and his supplier to ensure that quality products are used meeting the minimum engineering specifications.

The understanding of the basic elementary chemistry of the applicable, proven nanotechnology used will ensure that risks are minimised. The testing of the basic bitumen stabilising agent and the influence of the nano-particle (the emulsifying agent or surfactant) on the engineering properties was performed with the specific objective to demonstrate the need for engineers to realise the importance of quality control during the manufacturing, ensuring that material compatible materials are utilised, with chemical properties that will guarantee that macro projects can successfully be implemented using new disruptive nanotechnologies by adhering to basic scientific principles.

The main objectives of the modification of an emulsifying stabilising agent are to:

- Ensure that a high strength chemical bond is established between the stabilising agent and the naturally available granular materials;
- Achieve a high level of hydrophobicity of each particle within the granular materials layer through a 3-dimensional consolidation of the nano-silane, covering the total of surface area of each particle, and
- Protect each particle within the granular material against the effect of water and neutralise the presence of any secondary minerals (e.g., clay) within the granular material by ensuring that a material compatible nano-silane (modified where necessary with a HCT nano-particle), is utilised based on the scientifically measured mineralogy of the granular material.

A material-compatible organofunctional nano-silane with an applicable HCT (when needed), is introduced into the stabilising agent that will ensure that the required chemical bonds are activated during the stabilising process. The quantity of the modifier to be added is a function of the surface area to be covered. It follows that the fine fractions (the percentage passing the 0.075 mm sieve size and the percentage of clay (less than 0.002 mm fraction) is of importance in the adjustment of the nano-silane percentages to be added to achieve the objectives successfully. It follows that the modification of the stabilising agent to optimally achieve the required engineering properties is not conducive to a "one solution fits all" product approach.

It is shown that materials of a very poor quality (unthinkable using traditional pavement engineering approaches to material utilisation) can be treated and stabilised to obtain the required engineering properties meeting engineering specifications in terms of compressive strengths, tensile strengths and durability in pavement layers, depending on the design traffic loading. This is possible with the application of scientific knowledge and applying high-quality nano-scale products, i.e., emulsifying agents (surfactants), material-compatible nano-silane products with the required chemical compositions together with

material-compatible HCT modifications when applicable. These disruptive nanotechnologies have been proven in laboratories, through Accelerated Pavement Tests (APT) [8,9] and in practice on several roads in southern Africa [13].

**Author Contributions:** G.J.J., under the directive of the Head of Department of Civil Engineering, W.J.v.S., has been leading the research into the provision of affordable road infrastructure at the faculty of Engineering, University of Pretoria. W.J.v.S. recognized the potential of nanotechnology solutions in the field of pavement engineering more than a decade ago. G.J.J., through involvement in the private sector and the support of road authorities, has been instrumental in the development of scientific principles, ensuring that implementation can be achieved and tested at a minimum risk. All authors have read and agreed to the published version of the manuscript.

**Funding:** This research received no external funding.

**Institutional Review Board Statement:** Not applicable.

**Informed Consent Statement:** Not applicable.

**Data Availability Statement:** Not applicable.

**Acknowledgments:** The support of GeoNANO Technologies (Pty) Ltd., 18 Davies road, Wychwood, Germiston 1401, South Africa, Tel.: +27844078489, Available online: www.geonano.co.za, accessed on 7 October 2021, info@geonano.co.za, in support of students in the Department of Civil Engineering, University of Pretoria, Pretoria, South Africa to test a wide variety of materials as part of final year projects and post-graduate theses, testing the various principles identified in this article, is acknowledged.

**Conflicts of Interest:** The authors declare no conflict of interest.

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
