# Peer review of "Engineering Properties of New-Age (Nano) Modified Emulsion (NME) Stabilised Naturally Available Granular Road Pavement Materials Explained Using Basic Chemistry"

_applsci, doi:10.3390/app11209699_

Round 1
Reviewer 1 Report
The topic of the paper is in line with trend in pavement engineering that promotes the use of materials classified as non-standard and marginal as constituent pavement layer materials. In the paper the basic chemistry that influences the characteristics and expected outputs applicable to the use of New-age Modified Emulsion stabilizing agents is presented.
The main part of the paper is very general, focusing on the chemistry and the connection between described chemical reactions and pavement engineering (mechanical) properties is not clear. It is hard to understand which pavement materials/layers are in the focus of the paper.
Section 4 that discusses the influence of the chemical characteristics of the emulsifying agent on the potential engineering properties is based on a single reference (“South African National Standards, SANS 4001-BT3, (2014). Civil Engineering Specifications Part BT3: Anionic bitumen road emulsion. Pretoria, South Africa, 2014.”) - more research/references should be added.
Consider reorganize section 5 as it is very hard to follow. From the pavement engineering point of view the emphasis should be placed on chemical reactions which are crucial for obtaining certain pavement engineering properties.
Reviewer 2 Report
- What is the character of the paper. It seems like a literature review but number of references and differences in selection of authors is very poor. What are the new findings resulting from the paper?
- The objective of the paper is very vague. What authors mean in the sentence: “This paper aims to give a basic description of some basic pre-dominant chemical characteristics of an emulsion stabilizing agent”. The objective should be precise.
- The article is difficult to flow. Why there is Introduction and then background? What is the difference between those sections? Both sections are too long and the sense of studies is lost in it.
- Naming the cationic or anionic emulsifying agent as a “nanotechnology” is not proper. The title of point 3 is not adequate because the products described in those part are not “nanotechnology” materials.
- What is the practical meaning and purpose of presenting “bond strength” ranking in Figure 4?
- Explain what do you mean under the term “New-age nano modified emulsion” explain more clearly the differences between NME and traditional emulsion
- The classification presented in Figure 10 is trivial. Beside CBR numerous other factors determine the possibilities of application material to a base course.
- Title of section 5.4 “Material compatible NME nanotechnologies making the impossible possible in pavement engineering” is not clear and contains a mistake.
- When you use a name of tests or properties (eg. ITS UCS, RCS, RTS) at first time use a whole name and reference to the test procedure.
- The conclusions should better correspond to the title of the paper and give a brief summary, which of the basic chemical (relations? Formulations?) mostly correspond to mechanical properties of natural aggregates stabilized with NME. In current form conclusions concerns different aspects which seems not to be precisely related with the title of the paper.
Author Response
lease see the attachment.

Reviewer 3 Report
Some Keywords could be refined
Round 2
Reviewer 1 Report
Authors addressed the comments.
Reviewer 2 Report
Authors made some corrections in the paper and responses but they do not match the previous remarks, especially point 1), point 2),point 3)
Very similar paper have been published by authors recently in applied science “Fundamental Principles Ensuring Successful Implementation of New-Age (Nano) Modified Emulsions (NME) for the Stabilisation of Naturally Available Materials in Pavement Engineering”. Some parts overlaps.
I do not see any new knowledge in the paper. Authors attempted to describe of “the effects of the chemistry involved in the additives and modifications used in the manufacturing of bitumen emulsion as a stabilizing agent.” But I cannot find any practical and CLEAR findings in this description.
Response on point 6 is not satisfactionary and it contains not a true. I don’t agree that “the phrase “nano” has developed among officials and practitioners, not realizing the simple fact that it refers to nothing more than a measurement.” It is a significant misuse. According to this approach any chemical reaction, like cement setting process can be named “nanotechnology”. Nanotechnology is quite new field of technical science which based on precisely manipulating atoms and molecules for fabrication of macroscale products. We cannot name cement, lime or emulgulators used in bituminous emulsion as nanotechnology practices, just because they have small dimensions. This way of thinking has a consequence in the whole paper.
The paper at current form is not precise. Too many digressions make impossible to flow the main thought. For example in linies 598 – 604 authors describe the content of silicone in earth, what is completely outside the topic of the paper. There are numerous similar examples in the paper
The flow of the paper is not clear. From the one side authors presents a very long background, then explain some elementary chemical reactions, then presents some laboratory tests. I do not see what authors want exactly to transfer to readers?
